# Spontaneous Development of Dental Dysplasia in Aged *Parp-1* Knockout Mice

**DOI:** 10.3390/cells8101157

**Published:** 2019-09-27

**Authors:** Hisako Fujihara, Tadashige Nozaki, Masahiro Tsutsumi, Mayu Isumi, Shinji Shimoda, Yoshiki Hamada, Mitsuko Masutani

**Affiliations:** 1Biochemistry Division, National Cancer Center Research Institute 5-1-1 Tsukiji, Chuo-ku, Tokyo 104-0045, Japan; fujihara-h@tsurumi-u.ac.jp (H.F.); nozaki@cc.osaka-dent.ac.jp (T.N.); 2Department of Oral and Maxillofacial Surgery, School of Dental Medicine, Tsurumi University 2-1-3 Tsurumi, Tsurumi-ku, Yokohama, Kanagawa 230-8501, Japan; hamada-y@tsurumi-u.ac.jp; 3Department of Pharmacology, Faculty of Dentistry, Osaka Dental University 8-1, Kuzuhahanazono-cho, Hirakata, Osaka 573-1121, Japan; 4Department of Pathology, Saiseikai Chuwa Hospital 323 Oaza Abe, Sakurai City, Nara 633-0054, Japan; mtutumi@chuwa-hp.jp; 5Department of Frontier Life Sciences, Graduate School of Biochemical Science, Nagasaki University 1-7-1 Sakamoto, Nagasaki 852-8588, Japan; bb55317036@ms.nagasaki-u.ac.jp; 6Department of Oral Anatomy-1, School of Dental Medicine, Tsurumi University 2-1-3 Tsurumi, Tsurumi-ku, Yokohama, Kanagawa 230-8501, Japan; shimoda-s@tsurumi-u.ac.jp; 7Division of Cellular Signaling, National Cancer Center Research Institute 5-1-1 Tsukiji, Chuo-ku, Tokyo 104-0045, Japan

**Keywords:** poly(ADP-ribose) polymerase-1, aged mice, knockout mice, dental dysplasia

## Abstract

Poly(ADP-ribose) polymerase (Parp)-1 catalyzes polyADP-ribosylation using NAD^+^ and is involved in the DNA damage response, genome stability, and transcription. In this study, we demonstrated that aged *Parp-1^−/−^* mouse incisors showed more frequent dental dysplasia in both ICR/129Sv mixed background and C57BL/6 strain compared to aged *Parp-1^+/+^* incisors, suggesting that Parp-1 deficiency could be involved in development of dental dysplasia at an advanced age. Computed tomography images confirmed that dental dysplasia was observed at significantly higher incidences in *Parp-1^−/−^* mice. The relative calcification levels of *Parp-1^−/−^* incisors were higher in both enamel and dentin (*p* < 0.05). Immunohistochemical analysis revealed (1) Parp-1 positivity in ameloblasts and odontoblasts in *Parp-1^+/+^* incisor, (2) weaker dentin sialoprotein positivity in dentin of *Parp-1^−/−^* incisor, and (3) bone sialoprotein positivity in dentin of *Parp-1^−/−^* incisor, suggesting ectopic osteogenic formation in dentin of *Parp-1^−/−^* incisor. These results indicate that Parp-1 deficiency promotes odontogenic failure in incisors at an advanced age. Parp-1 deficiency did not affect dentinogenesis during the development of mice, suggesting that Parp-1 is not essential in dentinogenesis during development but is possibly involved in the regulation of continuous dentinogenesis in the incisors at an advanced age.

## 1. Introduction

Tooth function is quite important in our daily life, and includes mastication, speech, aesthetics, and facial and jaw development during growth periods. Teeth consist of mainly four elements—enamel, dentin, pulp, and cementum—and their development requires multilineage cells controlled by different signaling pathways, as in the development of other organs. During tooth development, both tooth formation and mineralization are necessary, and reciprocal interactions between dental epithelium and neural crest-derived ectomesenchyme (EM) are also required [1]. Briefly, dental epithelium develops both external and internal enamel epithelium from which ameloblasts differentiate, whereas EM cells differentiate into odontoblasts [2]. Ameloblasts are responsible for secretion of enamel matrix, which is a highly mineralized tissue and the hardest tissue in the body [3,4]. Meanwhile, odontoblasts are responsible for dentin formation, and dental pulp is the soft tissue inside the tooth that supports dentin formation and regeneration [5,6]. Moreover, dentin should be formed before enamel formation. After crown formation is completed, both external and internal enamel epithelial cells proliferate and form Hertwig’s epithelial root sheath, which is necessary for root formation [7]. Then, it induces differentiation of EM cells from radicular pulp into odontoblasts that form root dentin. The mechanism of crown and root formation, and related transcription factors and growth factors, have been reported [2,3,4].

The process of dentinogenesis, including secretion of extracellular matrix (ECM) proteins and regulation of dentin mineralization, is highly controlled by odontoblasts. Dentin sialophosphoprotein (DSPP) is the most abundant ECM in dentin and is processed into dentin sialoprotein (DSP), dentin glycoprotein (DGP), and dentin phosphoprotein (DPP) [8,9,10]. Among them, DSP and DPP are mainly expressed in odontoblasts and dentin [11,12]. On the other hand, production of enamel by ameloblasts is regulated by laminin, amelogenin, and type I collagen [3], in addition to reciprocal signaling interactions between dental epithelium and mesenchyme [13,14,15].

To achieve the above-mentioned complicated dentinogenesis, regulation of differentiation by chromatin factors is expected to be important, and poly(ADP-ribose) polymerase-1 (Parp-1) could be one of such factors. PARP-1 is the most abundant PARP enzyme in the Parp family, which catalyzes poly(ADP-ribosyl)ation consuming nicotinamide adenine dinucleotide (NAD^+^) as a substrate to target proteins that lead to biological activities. Poly(ADP-ribosyl)ation is known to be involved in many cellular processes, including transcriptional regulation and differentiation, as well as DNA repair [16,17], cell death [18], telomere regulation [19], and genomic stability [20]. Moreover, its deletion was reported to lead to increased sensitivity to anticancer drugs and ionizing radiation in mice [20,21]. In addition to these biological activities, accumulating studies have shown Parp-1 involvement in differentiation processes. For example, trophoblast differentiation was shown to be regulated by Parp-1 [22]. Parp inhibitors caused differentiation of blood cancer cells [23]. Parp-1 knockout mice at an advanced age showed augmented incidences of tumor development [24] and showed increased frequency of deletion-type mutations in the liver [25]. We have also recently reported that a Parp inhibitor (PJ34) suppressed osteogenic differentiation of mouse mesenchymal stem cells in vitro via modulation of the BMP-2 signaling pathway [26]. 

In this study, we surveyed histopathological changes in the entire body of *Parp-1^−/−^* mice at an advanced age, and unexpectedly observed a higher incidence of dental dysplasia in the incisors of *Parp-1^−/−^* mice compared with those of *Parp-1^+/+^* mice, and also analyzed them using micro-CT.

## 2. Materials and Methods

### 2.1. Animal Experiments

*Parp-1^−/−^* mice were previously generated by disrupting the Parp-1 exon 1 through the insertion of a neomycin resistance gene cassette [21]. Both *Parp-1^+/+^* and *Parp-1^−/−^* male mice of two strains were produced—a mixed genetic background of ICR/129SV and a mixed genetic background of C57BL/6 by line-breeding, respectively. They were housed five each in plastic cages under specific pathogen-free conditions, with a 12-h light–dark cycle until they were analyzed. The mice were continuously supplied with normal food chow (CE-2; CLEA Japan, Inc., Tokyo, Japan) and sterilized water ad libitum.

When they became 110 weeks old, they were sacrificed by cervical dislocation. Consequently, they were fixed in 10% formalin/phosphate buffered saline (PBS, Wako Pure Chemical Industries, Ltd., Osaka, Japan) for 48 h. All animal studies were approved by the Animal Experimental Committee of the National Cancer Center and performed following the Guidelines for Animal Experiments of the National Cancer Center, which meet the ethical guidelines for experimental animals in Japan. The numbers of Ethical Board Approval of National Cancer Center Research Institute are as follows; T03-C0047, T03-C0048, T03-C0051, T03-C0056, T03-C0058, and T03-C0061. 

### 2.2. High Resolution Radiography of Incisors

To obtain high resolution radiographs of mouse incisors, each maxilla was analyzed after formalin fixation followed by hand-removal of surrounding tissue. Each maxilla was X-rayed on a 3D Measuring X-Ray Computed Tomography (CT) Scanner (TDM1000, Yamato Scientific Co. Ltd., Tokyo, Japan). The setting of the CT was as follows; tube voltage was 90 kV, tube current was 19 µA, tomography interval was 9.6 µm, figure measurement was 9.6 mm, matrix size was 1000 × 1000 × 1000, and integration for 16 times. Then, all data were analyzed by 3D Trabecular Bone Structure Measurement Software (TRI/3D-BON, Ratoc Company, Tokyo, Japan). Evaluation items were as follows; (1) volume of teeth, (2) volume of pulp cavity, (3) pulp thickness, (4) relative evaluation of calcification levels of enamel, (5) dentin, (6) pulp, (7) dentin + pulp, and (8) the whole of the teeth of the coronal section at the middle of sinus (air was set to be zero reference), respectively. The areas of enamel, dentin, pulp and the whole tooth for each evaluation were defined in Appendix A. Moreover, the calcification level of dentin was defined as the average of four points; ① medial, ② distal, ③ labial, and ④ palatal points of dentin, respectively. Six *Parp-1^+/+^* and ten *Parp-1^−/−^* mice (C57BL/6) were used in this analysis.

### 2.3. Decalcification

After radiographical analysis was completed, the maxilla of each mouse was decalcified in OSTEOSOFT (Merck KGaA, Darmstadt, Germany) at 4 °C by changing the liquid every three days for four weeks [27] and formic acid decalcification was also used for other cases.

### 2.4. Histopathological Analysis 

The maxilla was immersed in 70% alcohol for 48 h, dehydrated, and embedded in paraffin. Samples were sectioned in the coronal plane for ICR/129Sv mice and sectioned in the sagittal plane for C57BL/6 mice. Tissue sections (4 µm) were mounted on silane-coated slides (New Silane III, Muto Pure Chemicals Co., LTD., Tokyo, Japan), deparaffinized with xylene (Wako Pure Chemical Industries, Ltd., Tokyo, Japan), and rehydrated with graded alcohol solutions (Wako Pure Chemical Industries, Ltd.). The specimens were pathologically analyzed by hematoxylin and eosin staining (HE; Merck KGaA, Darmstadt, Germany).

### 2.5. Immunohistochemistry

Tissue sections mounted on slides were deparaffinized with xylene and rehydrated with descending concentrations of ethanol. Antigen retrieval was then performed using Immunosaver (Nisshin EM, Tokyo, Japan) according to the manufacturer’s protocol. Briefly, sections were incubated in Immunosaver (1:200 dilution in tap water) for 40 min at 95 °C and then transferred to tap water and incubated for 10 min at room temperature. Subsequently, endogenous peroxidase was inactivated by treatment with 3% hydrogen peroxide (Wako Pure Chemical Industries, Ltd.) in methanol (Wako Pure Chemical Industries, Ltd.) for 30 min at room temperature [28].

Antibodies used in this study were rabbit polyclonal anti-bone sialoprotein (BSP) antibody (1:200 dilution, Rockland, Leimrick, PA, USA), monoclonal anti-dentin sialoprotein (DSP) antibody (1:250 dilution, Merk Millipore, Kenilworth, NJ, USA), and monoclonal anti-PARP-1 antibody (1:50 dilution, Santa Cruz Biotechnology, Inc., Santa Cruz, CA, USA). 

For immunohistochemistry of anti-BSP antibody, slides were treated with 20% normal goat serum (Nichirei Corporation, Tokyo, Japan) for 30 min at room temperature and were incubated with the primary antibody at 4 °C overnight. Antibodies were diluted with PBS (pH 7.4) containing 1% bovine serum albumin (Sigma-Aldrich, St. Louis, MO, USA) and incubated at 4 °C. After several washes with PBS, bound antibodies were visualized using Histofine Simple Stain MAX-PO(MULTI) (Nichirei, Tokyo, Japan) and 3,3′-diaminobenzidine (Vector Laboratories, Burlingame, CA, USA) according to the manufacturer’s protocol. 

For immunohistochemistry of anti-DSP antibody and anti-PARP-1 antibody, Mouse on Mouse (M.O.M.^®^) Basic Kit and Vectastain Elite^®^ ABC kit (both from Vector Laboratories, Burlingame, CA, USA) were used according to the manufacturer’s protocol.

The sections were counterstained with hematoxylin and mounted. As a negative control, sections were processed without exposure to the primary antibody. The sections were viewed under BX51 System Microscope (Olympus, Tokyo, Japan). Images were recorded using a digital microscope camera (DP70; Olympus). Four *Parp-1^+/+^* and four *Parp-1^−/−^* mice were used in this analysis.

### 2.6. Semi-Quantitative Analysis of Immunohistochemical Stainings

The photographs of the immunohistochemical analysis with each antibody were analyzed by ImageJ Fiji (National Institutes of Health, Bethesda, MD, USA, available form https://imagej.net/Fiji/Downloads) and mean intensity of selected area was calculated. First, all photographs were deconvoluted and DAB color version images were chosen from divided three images. Then, subjected area was selected and cropped; (1) odontoblasts lineage for Parp-1 positivity, (2) ameloblasts lineage for Parp-1 positivity, (3) dentin area for DSP positivity, and (4) dentin area for BSP positivity. Subsequently, mean density of DAB color was calculated and the number was converted from max intensity of 255 [29]. The representative immunostaining data for Figures 7–9 of each genotype were used in this analysis.

### 2.7. Statistical Analysis

All statistical analyses were performed with EZR (Saitama Medical Center, Jichi Medical University, Saitama, Japan), which is a graphical user interface for R (The R Foundation for Statistical Computing, Vienna, Austria). More precisely, it is a modified version of R Commander designed to add statistical functions frequently used in biostatistics [30]. In detail, chi-square test was performed for the analysis of the incidence of dysplasia and denticle in incisors of both genotypes (Table 1 and Table 2). Students’ *t*-test was performed for the analysis of general structure of incisors of both genotypes.

## 3. Results

### 3.1. Higher Incidence of Dental Dysplasia in Aged Parp-1^−/−^ Incisors of an ICR/129Sv Mixed Genetic Background

Comparison of histopathological changes in aged *Parp-1^+/+^* and *Parp-1^−/−^* mice at 110 weeks old of an ICR/129Sv mixed genetic background were carried out with microscopic analysis. We found the development of dental dysplasia in incisors but not in molars or other teeth of aged *Parp-1^+/+^* and *Parp-1^−/−^* mice. Coronal sections of incisors of most of the aged *Parp-1^+/+^* mice showed the tooth structure of enamel, dentin, predentin, and pulp, similar to those of mature mice, resembling the human tooth. Moreover, a layer of odontoblasts was observed in the periphery of dental pulp (Figure 1a,c). However, coronal sections of incisors of 110-week-old *Parp-1^−/−^* mice frequently showed failure to form a normal tooth structure, especially in the lower half area of the sections. Dental dysplasia, such as hypertrophy, distortion of tooth architecture, and several denticle structures, was observed inside the aged *Parp-1^−/−^* incisors (Figure 1b,d).

In aged ICR/129Sv mice, four out of 13 *Parp-1^+/+^* incisors and all of the aged *Parp-1^−/−^* incisors showed dental dysplasia, respectively. Denticles were also observed in four out of 13 *Parp-1^+/+^* and eight out of nine *Parp-1^−/−^* incisors (Table 1). All of the *Parp-1^−/−^* incisors that showed dental dysplasia harbored denticles. To analyze whether increased development of dental dysplasia and denticles in *Parp-1^−/−^* incisors also appear in younger animals, we analyzed incisors at 41-weeks-old (data not shown), however, dental dysplasia and denticles were not observed in either *Parp-1^+/+^* or *Parp-1^−/−^* incisors, suggesting that only aged incisors show an increased incidence of dental dysplasia under *Parp-1* deficiency.

### 3.2. Histopathological Analysis of Incisors of Aged Parp-1^+/+^ and Parp-1^−/−^ Mice (C57BL/6)

Next, to clarify the effect of genetic background for the phenomena, we analyzed development of dental dysplasia in a different genetic background using line breeding with C57BL/6 mouse strain. *Parp-1^+/−^* mice backcrossed to C57BL/6 strain were used to generate *Parp-1^+/+^* and *Parp-1^−/−^* mice. Microscopic analysis of sagittal incisor sections of 110-week-old *Parp-1^+/+^* mice showed a clear and distinct tooth structure of enamel, dentin, predentin, and pulp, consistent with ICR/129Sv mouse incisors. The layer of odontoblasts was also observed in the dental pulp periphery (Figure 2a,c,f) and the layers of ameloblasts were also apparent from apex to incisor edge (Figure 2a,c,d). On the other hand, the incisors of *Parp-1^−/−^* counterparts showed a failure to form a normal tooth structure in the sagittal sections. Inside the incisors, dental dysplasia, such as ectopic and hypertrophic dentin formation, was also observed, and accordingly, the layer of odontoblasts at the margin of the pulp had failed and the alteration of cellular morphology of odontoblasts was also observed (Figure 2b,g,j). The layers of ameloblasts were also affected in *Parp-1^−/−^* mice incisors. These showed almost normal ameloblast layers from the apex to one-third from the apex, however, the height of ameloblasts of the medial half of the incisors became shorter with a shrunken cellular morphology (Figure 2b,g,h). Most of the structure of the enamel disappeared by the decalcification process in both *Parp-1^+/+^* and *Parp-1^−/−^* mice incisors (Figure 2a,b). Cementum of *Parp-1^+/+^* and *Parp-1^−/−^* mouse incisors was not detectable, therefore we performed immunohistochemical analysis of BSP in the later section as a biomarker of cementum and periodontal ligament. From HE staining and radiographic analysis, the incidences of dental dysplasia of the aged, mixed genetic background of C57BL/6 were also evaluated. As shown in Table 2, two out of six *Parp-1^+/+^* mice and nine out of ten aged *Parp-1^−/−^* mice showed dental dysplasia, respectively. Moreover, denticle structures were observed coexisting with dysplasia in four out of ten *Parp-1^−/−^* mice, but not in *Parp-1^+/+^* mice. 

Therefore, the incidences of dental dysplasia and denticle structure in the incisors were significantly higher in both ICR/129Sv and C57BL/6 *Parp-1^−/−^* mice compared with those of *Parp-1^+/+^* mice, respectively (*p* < 0.01).

### 3.3. Radiological Analysis Showed Dental Dysplasia and Denticle Structures in Aged C57BL/6 Parp-1^−/−^ Incisors

The incisors of aged *Parp-1*^+/+^ mice showed a normal structure of a tooth, including enamel, dentin, and pulp. Moreover, the boundary of the dentin and pulp cavity was smooth and constant (Figure 3a,c,e,g). On the other hand, incisors of aged *Parp-1^−/−^* mice showed an abnormal form of dentin and pulp cavity structure, widely expanded form at the upper area and shrunken form at the bottom area in the coronal plane (Figure 3b). The boundary of dentin and pulp cavity was bumpy and uneven, and ectopic dentin-like structures and denticle structures were observed inside the pulp cavity in all planes (Figure 3b,d,f,h).

### 3.4. Three-Dimensional Analysis Based on Radiological CT Images in Aged C57BL/6 Parp-1^−/−^ Incisors 

Three-dimensional figures of both *Parp-1^+/+^* and *Parp-1^−/−^* maxillae were constructed, based on the radiological analysis of CT images. The boundary of the dentin and pulp cavity of incisors of *Parp-1^+/+^* mice was smooth and constant (Figure 4a). On the other hand, incisors of the *Parp-1^−/−^* counterparts showed an abnormal form of dentin and pulp cavity structure, uneven boundary of dentin and pulp cavity, and ectopic dentin and denticle structures, respectively (Figure 4b).

To evaluate the incisor structure itself, another type of three-dimensional analysis was performed. The pulp cavity was drawn in orange and hard tissue such as enamel and dentin was drawn in yellow (Figure 4c,d). The pulp cavity of *Parp-1^+/+^* mouse was wide at the apex of the incisors, gradually became narrower, and disappeared at the incisor margin (Figure 4c). On the other hand, the pulp cavity of *Parp-1^−/−^* mouse was also wide at the apex, it was widely spread inside the incisors, and did not show any narrowing structures (Figure 4d).

### 3.5. Tooth and Pulp Volumes of Aged Parp-1^−/−^ Incisors Based on 3D Radiological CT Images 

To evaluate how ectopic dentin and denticle structures in the *Parp-1^+/+^* and *Parp-1^−/−^* incisors in the C57BL/6 strain affected tooth and pulp cavity volume and pulp thickness, these were calculated using 3D Trabecular Bone Structure Measurement Software (TRI/3D-BON, Ratoc Company, Tokyo, Japan).

The average tooth volume of aged *Parp-1^+/+^* and *Parp-1^−/−^* incisors was 6.85 mm^3^ (range; 5.51–8.65, S.D. = 1.50) and 6.90 mm^3^ (range; 5.45–9.67, S.D. = 1.86), respectively. The average pulp cavity volume of aged *Parp-1^+/+^* and *Parp-1^−/−^* incisors was 1.92 mm^3^ (range; 1.05–3.88, S.D. = 1.05) and 1.28 mm^3^ (range; 0.63–2.11, S.D. = 0.71), respectively. There was no significant difference between the genotypes in either tooth or pulp volume, however, there was a tendency for smaller pulp cavity in *Parp-1^−/−^* incisors (*p* = 0.181, Figure 5a). The average pulp thickness of aged *Parp-1^+/+^* and *Parp-1^−/−^* incisors was 336.4 µm (range; 267.8–411.9, S.D. = 87.6) and 316.3 µm (range; 199.2–396.4, S.D. = 63.8), respectively. There was no significant difference between the genotypes in the pulp thickness (Figure 5b).

### 3.6. Relative Calcification Levels of Each Element of Tooth were Significantly Higher in Aged Parp-1^−/−^ Incisors

To evaluate the function of ameloblasts and odontoblasts in aged *Parp-1^+/+^* and *Parp-1^−/−^* incisors in the mixed genetic background of C57BL/6, relative calcification levels of each element of a tooth were calculated using the same software, 3D Trabecular Bone Structure Measurement Software. Air was set to be the zero reference. Six *Parp-1^+/+^* mice and ten *Parp-1^−/−^* mice were used in this analysis.

The relative calcification levels of enamel were 1939.6 (range; 1077.8–2459.0, S.D. = 447.1) and 2173.0 (range; 1049.5–2788.0, S.D. = 610.2), that of dentin were 1243.1 (range; 922.0–1461.8, S.D. = 175.5) and 1401.8 (range; 1265.2–1527.7, S.D. = 79.6), that of pulp were 321.7 (range; 231.7–410.1, S.D. = 58.4) and 348.2 (range; 176.0–886.9, S.D. = 156.3), that of dentin + pulp were 640.3 (range; 490.4–875.4, S.D. = 118.8) and 814.6 (range; 532.1–1096.0, S.D. = 192.0), and that of a total tooth were 675.5 (range; 475.6–897.7, S.D. = 134.1) and 866.0 (range; 584.2–1123.2, S.D. = 199.9) in aged *Parp-1^+/+^* and *Parp-1^−/−^* mice, respectively. In these elements of teeth, aged *Parp-1^−/−^* incisors showed significantly higher calcification levels in enamel, dentin, dentin + pulp, and total of a tooth (*p* < 0.05) (Figure 6a–e).

### 3.7. Immunohistochemical Analysis of Parp-1 in Parp-1^+/+^ and Parp-1^−/−^ Incisors

To confirm that the failure of odontoblast and ameloblast layers or alteration of cellular morphology were related to the lack of Parp-1, the expression of Parp-1 protein was analyzed by immunohistochemical analysis using four mice of each genotype. *Parp-1^+/+^* incisors showed positive Parp-1 immunostaining in ameloblast (Figure 7c,h) and odontoblast (Figure 7f,g) layers and their localization was mostly in cytoplasm, but occasionally in nuclei (Figure 7a,c–h). The long term decalcification process may have caused the preferential loss of antigen detection in nuclei compared with cytoplasm. Parp-1 was not detected at the apex area and inside of the pulp of *Parp-1^+/+^* incisor (Figure 7e, f) and Parp-1 was absent in any elements of *Parp-1^−/−^* incisor (Figure 7b,i–n).

To quantify the intensity of DAB staining for Parp-1 staining, the photos of both *Parp-1^+/+^* and *Parp-1^−/−^* incisors represented in Figure 7 were analyzed by ImageJ Fiji. The average value of relative intensity of DAB staining of Parp-1 was 2.95 folds higher in odontoblast area (Figure 7g,m, and the legend) and 1.83 folds higher in ameloblast area compared with *Parp-1^+/+^* incisor (Figure 7h,n, and the legend), respectively.

### 3.8. Immunochemical Analysis of Dentin Sialoprotein (DSP) and Bone Sialoprotein (BSP) in Parp-1^+/+^ and Parp-1^−/−^ Incisors

Next, expression and localization of dentin sialoprotein (DSP), a biomarker of dentin, and bone sialoprotein (BSP), a biomarker of cementum and periodontal ligament, were analyzed using an immunohistochemical method in the mixed genetic background of C57BL/6. Although we used serial sections of each sample, direct merging of different immunohistochemical samples was not possible. Strong expression of DSP was observed in the dentin area of aged *Parp-1^+/+^* incisor (Figure 8a,d,f–h), whereas weaker expression of DSP was sparsely observed inside aged *Parp-1^−/−^* incisor, which could be considered as dentin area (Figure 8b,i,j,l–n).

To quantify the intensity of DAB staining for DSP staining in dentin area, the photos of both *Parp-1^+/+^* and *Parp-1^−/−^* incisors represented in Figure 8 were analyzed by ImageJ Fiji. The average value of relative intensity of DSP positivity in dentin area was 2.38 folds higher in *Parp-1^+/+^* incisor compared with *Parp-1^−/−^* incisor (Figure 8g,h,m,n, and the legend). Moreover, dentinal tubules were clearly observed in dentin area of *Parp-1^+/+^* incisor (Figure 8g), however, they were hardly observed in *Parp-1^−/−^* incisor (Figure 8m,n).

From the result of DSP staining, dentin formation inside aged *Parp-1^−/−^* incisors was considered ectopic and not fully matured.

On the other hand, the expression of BSP was observed in periodontal ligament of both aged *Parp-1^+/+^* (Figure 9a,f–h) and *Parp-1^−/−^* incisors (Figure 9b,l,n) but the level was weaker in *Parp-1^−/−^* incisor. BSP was also observed in cementum of *Parp-1^+/+^* incisor (Figure 9g,h), but was hardly detectable in *Parp-1^−/−^* incisor (Figure 9n). BSP was not detected at the apex area of neither *Parp-1^+/+^* and *Parp-1^−/−^* incisors nor in the dentin area of *Parp-1^+/+^* incisor, however, a scattered expression pattern of BSP was observed in predentin, especially in the severe dental dysplasia area of *Parp-1^−/−^* incisor (Figure 9j,m; red arrows). To quantify the intensity of DAB staining for BSP staining in dentin area, the photos of both *Parp-1^+/+^* and *Parp-1^−/−^* incisors represented in Figure 9 were analyzed by ImageJ Fiji. The average value of relative intensity of BSP positivity in dentin area was 2.54 folds higher in *Parp-1^−/−^* incisor compared with *Parp-1^+/+^* incisor (Figure 9g, m, and the legend). Moreover, BSP-positive cells were observed both in pulp cavity and dentin areas in *Parp-1^−/−^* incisor (Figure 9j,m). These BSP-positive dentin area and pulp cells of *Parp-1^−/−^* incisor could be possible evidence of ectopic bone-like formation in dentin and osteoblast-like cells in pulp in aged *Parp-1^−/−^* incisor.

## 4. Discussion

In this study, aged *Parp-1^−/−^* mouse incisors showed more frequent dental dysplasia and denticle formation in both ICR/129Sv mixed background and C57BL/6 strain, while aged *Parp-1^+/+^* incisors showed only slight changes in both genetic backgrounds (Table 1 and Table 2), though, dental dysplasia was not observed in younger mice or in molars or other teeth of aged mice in either genotype. Considering these results, Parp-1 would not be essential in dentinogenesis during development, but is possibly involved in the regulation of continuous dentinogenesis in the incisors at an advanced age.

We analyzed aged *Parp-1^+/+^* and *Parp-1^−/−^* incisors of both ICR/129Sv mixed genetic background and C57BL/6 strain histopathologically, but immunohistochemical analysis for incisors of the ICR/129Sv mixed genetic background was not successful using anti-Parp-1, DSP, and BSP antibodies, possibly because the incisor samples of the ICR/129Sv mixed genetic background were relatively older and not in good condition after long-term decalcification. We found that expression of Parp-1 was detected in the ameloblasts and odontoblasts in *Parp-1^+/+^* incisor, whereas Parp-1 protein expression was not detected anywhere in *Parp-1^−/−^* incisor (Figure 7b,i–n), as expected. The role of poly(ADP-ribosyl)ation during tooth development, including differentiation of ameloblasts and odontoblasts, enamel and dentin formation, and cementum and periodontal ligament formation has not been reported, and no aberration during tooth development was observed for exon1-, exon2-, and exon4-disrupted *Parp-1* knockout mice [21]. However, it is reasonable to hypothesize that differentiation and/or proliferation of ameloblasts and odontoblasts could be regulated by poly(ADP-ribosyl)ation, considering that poly(ADP-ribosyl)ation is known to be involved in many cellular processes, including differentiation of osteoblasts [26].

To confirm the effect of Parp-1 deficiency on tooth development in aged *Parp-1^−/−^* incisors, immunohistochemical analysis of DSP and BSP was performed. Expression patterns of DSP and BSP were different between *Parp-1^+/+^* and *Parp-1^−/−^* incisors, respectively. Briefly, DSP was strongly positive in the dentin area in *Parp-1^+/+^* incisor, while DSP positive areas were weaker and sparsely observed in *Parp-1^−/−^* incisor. The expression of BSP was detected in the cementum area of *Parp-1^+/+^* incisor but not in *Parp-1^−/−^* incisor. On the other hand, BSP expression was detected in the dentin area of *Parp-1^−/−^* incisor, especially in severe dental dysplasia showing cell-like pattern, which suggested possible ectopic osteogenesis in the dentin area, whereas it was negative in dentin of *Parp-1^+/+^* incisor. Therefore, the observed DSP and BSP detection pattern suggests that Parp-1 deficiency is likely to cause failure of precise dentinogenesis in incisors at an advanced age. In addition to the results of immunohistochemical analysis, CT images showed significantly higher relative calcification levels in both enamel and dentin in *Parp-1^−/−^* mice (*p* < 0.05, Figure 6). It was not possible to analyze enamel or enamel-related protein histopathologically, because it was lost by decalcification, which was required for preparation of paraffin sections. It was notable that Parp-1 expression was detected in arrays of ameloblasts and odontoblasts in *Parp-1^+/+^* incisor, and relative calcification levels were significantly higher in *Parp-1^−/−^* incisor, respectively.

Unexpectedly, the parameters of the general structure of incisors at an advanced age, including volume of the tooth, that of pulp cavity, and thickness of pulp, did not exhibit statistically significant differences between either genotype, and only pulp cavity showed a tendency to be smaller in *Parp-1^−/−^* incisors. The reason why the parameters of the general structure did not show significant difference could be the wide range of morphological variations in incisors of *Parp-1^−/−^* mice; *Parp-1^−/−^* incisors showed varied thickness of pulp based on the diverse shapes of pulp cavities, for example, some of them showed almost a closed pulp cavity because of the denticle structure inside of the incisors. As a result of this, pulp thickness seems to be larger in *Parp-1^−/−^* incisors (Figure 3 and Figure 4), however, there was a tendency for a smaller pulp cavity in *Parp-1^−/−^* incisors (*p* = 0.181, Figure 5a). In this study, only male mice were analyzed because it was preferred to analyze the effect of Parp-1 deficiency on dental differentiation at an advanced age in the absence of the effect of the estrous cycle. However, a gender difference of tooth formation has been reported using *Trps1^+/−^* (trichorhinophalangeal syndrome heterozygous) mice showing significantly smaller crown and root volumes in female *Trps1^+/−^* mice compared with male *Trps1^+/−^* mice [31]. Trichorhinophalangeal syndrome is a skeletal dysplasia with skeletal defects as well as dental abnormalities, where *Trps1* gene regulates dental mineralization. Although *Parp-1* deficiency has not been reported to be associated with such autosomal dominant genetic disorders, it could be possible that aged female *Parp-1^−/−^* incisors might show a difference from male ones in their general structure, considering that dental mineralization could be affected by gender. Therefore, possible gender differences should be studied further. From the view point of skeletal structure, particular body growth inhibition was not observed in *Parp-1^−/−^* mice from body weight comparisons between the two genotypes (data not shown). The eating behavior might not be affected by *Parp-1* deficiency, although *Parp-1^−/−^* mice in the C57BL6 strain showed a slightly lower bodyweight from a young age as previously reported [32].

For an interim summary, it is suggested that the activity of Parp-1 in odontoblasts and ameloblasts could be a direct key factor of dentinogenesis in an aged condition, based on our results. Parp-1 is known as a nuclear protein, as well as cytoplasmic protein [33], and it is occasionally detected in the cytoplasm in tissue staining [34]. We consider that Parp-1 activity either in nuclei or cytoplasm of *Parp-1^+/+^* mice could contribute as a possible differentiation regulator for odontoblasts and ameloblasts. Therefore, the possible mechanism of Parp-1 involvement in odontogenic development of incisors at an advanced age could be that (1) Parp-1 might be directly involved in regulation of DSP expression in odontoblasts; (2) the loss of Parp-1 expression might cause a failure in differentiation from ectomesenchymal cells into odontoblasts in *Parp-1^−/−^* incisors, which resulted in increased differentiation into osteoblasts; (3) the layer formation of odontoblasts was also affected by the loss of Parp-1 expression. As a result, relative calcification levels of enamel and dentin would become significantly higher in *Parp-1^−/−^* incisors.

There are other possible ways of indirect Parp-1 involvement in odontogenic differentiation. Considering the three facts that (1) rodents’ incisors continuously grow throughout their lives, (2) Parp-1 deficiency is reported to accelerate senescence by causing genomic and epigenetic instabilities [35], and (3) dentin and cementum are reported to be possible factors to estimate age [36], Parp-1 may accelerate senescence, which affects odontogenic differentiation by losing the capacity for renewal and regeneration.

To further understand how Parp-1 activity is involved in dentinogenesis/odontogenesis during aging, various possible interacting factors, which are also related to dentinogenesis, must be investigated—Wnt [37,38], hedgehog [39,40], Notch [41], and TGF-β signaling [42]—in relation to poly(ADP-ribosyl)ation.

As a clinical importance, ectopic osteogenic formation in dentin has been reported in patients with osteogensis imperfecta type I, III, IV with mutation of *COLIA2* (MIM #120160) [43], and VII [43,44]. When it comes to more familiar clinical problems, patients with dental dysplasia and root canal narrowing would definitely result in difficult root canal treatment for dentists and its success ratio would decrease [45], followed by apical periodontitis, which causes pain, swelling, pus discharge, and so on [46]. Therefore, if dental dysplasia in *Parp-1^−/−^* incisors could be a model of odontogenesis imperfecta, dentinogenesis imperfect, or dental dysplsia, the analysis of PARP-1 aberration would be a big key in looking for insights into not only these inherited autosomal dominant diseases, but also treatment of root canal.

## 5. Conclusions

In this study, aged *Parp-1^−/−^* mouse incisors showed frequent dental dysplasia and denticle formation in both ICR/129Sv mixed background and C57BL/6 strain, suggesting that Parp-1 deficiency itself could be involved in development of dental dysplasia at an advanced age. Although Parp-1 is not essential in dentinogenesis during development, Parp-1 is possibly involved in the regulation of continuous dentinogenesis in the incisors at an advanced age.

## Figures and Tables

**Figure 1 cells-08-01157-f001:**
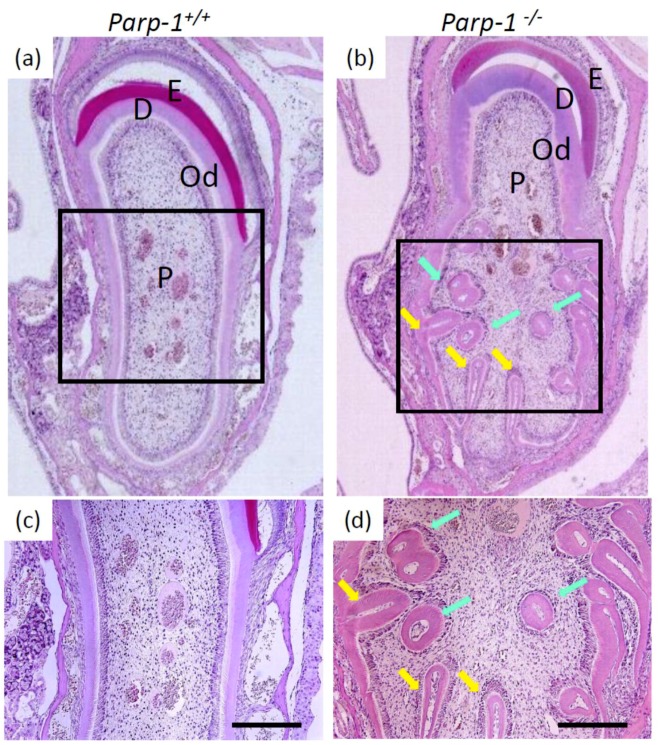
Histopathological analysis of incisors of aged *Parp-1^+/+^* and *Parp-1^−/−^* mice (ICR/129Sv). (**a**) hematoxylin and eosin staining (HE) staining of coronal plane of *Parp-1^+/+^* incisor. (**b**) HE staining of coronal plane of *Parp-1^−/−^* incisor. (**c**) and (**d**) are a higher magnification of the black box of (a) and (b), respectively. Light blue arrows indicate denticle structure and yellow arrows indicate dental dysplasia. E = enamel, D = dentin, Od = odontoblast, P = pulp. Bars indicate 200 µm.

**Figure 2 cells-08-01157-f002:**
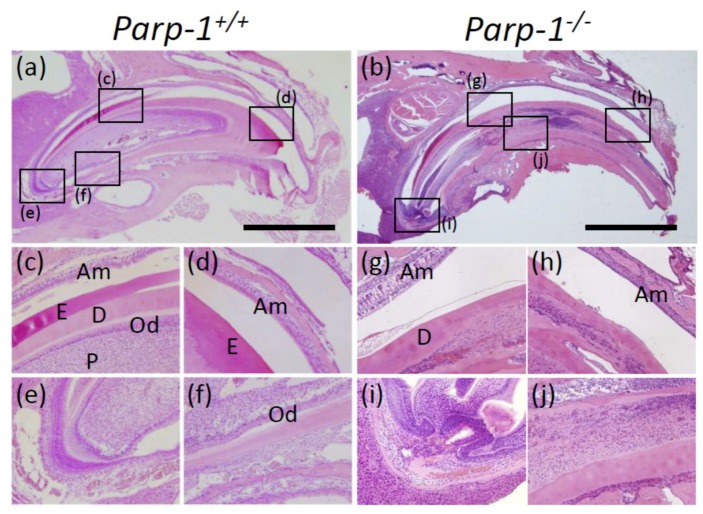
Histopathological analysis of incisors of aged *Parp-1^+/+^* and *Parp-1^−/−^* mice (C57BL/6). HE staining of sagittal plane of *Parp-1^+/+^* (**a**) and *Parp-1^−/−^* incisors (**b**). (**c–f**): A higher magnification of *Parp-1^+/+^* incisor showing the distinct structure of normal tooth. On the other hand, a higher magnification of the *Parp-1^−/−^* incisor showed almost normal ameloblast layers 1/3 from the apex (**g**), however, it turned to layers of shrunken ameloblast at the incisor edge (**h**). Odontoblasts of the *Parp-1^−/−^* incisor were almost normal in the apex area (**i**) and they also turned to shrunken cells and moreover, the layer itself disappeared (**j**). Am, ameloblast; Od, odontoblast; E, enamel; D, dentin; P, pulp. Bars indicate 2 mm.

**Figure 3 cells-08-01157-f003:**
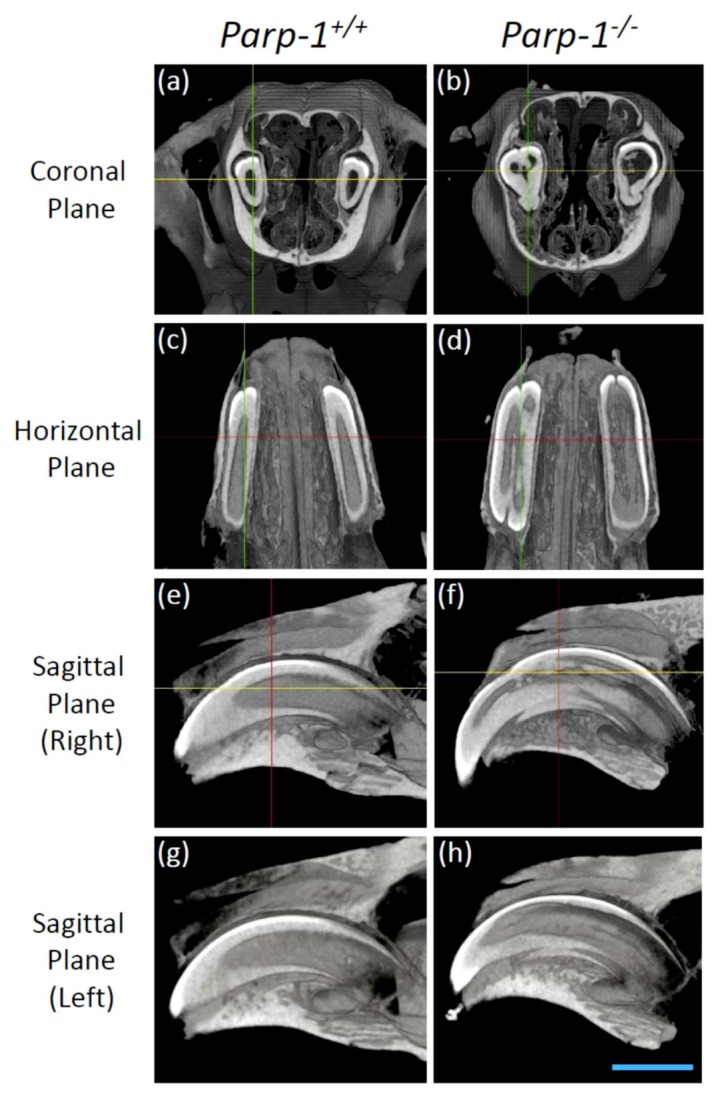
Representative CT images of incisors of aged *Parp-1^+/+^* and *Parp-1^−/−^* mice (C57BL/6). Mice incisors were analyzed from the coronal, horizontal, and sagittal plane, respectively. (**a**) Coronal plane of *Parp-1^+/+^* mouse. (**b**) Coronal plane of *Parp-1^−/−^* mouse. (**c**) Horizontal plane of *Parp-1^+/+^* mouse. (**d**) Horizontal plane of *Parp-1^−/−^* mouse. (**e**) Sagittal plane of right incisor of *Parp-1^+/+^* mouse. (**f**) Sagittal plane of right incisor of *Parp-1^−/−^* mouse. (**g**) Sagittal plane of left incisor of *Parp-1^+/+^* mouse. (**h**) Sagittal plane of left incisor of *Parp-1^−/−^* mouse. Bar indicates 2 mm.

**Figure 4 cells-08-01157-f004:**
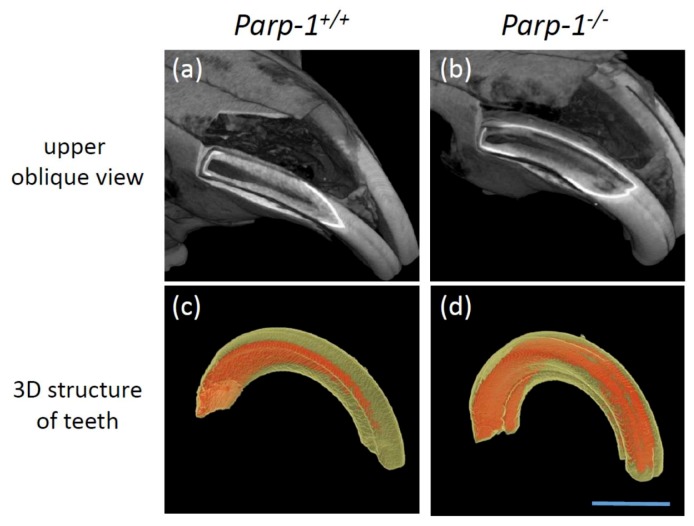
(**a**) Upper oblique view of *Parp-1^+/+^* mouse and (**b**) upper oblique view of *Parp-1^−/−^* mouse via 3D images. (**c**) Three-dimensional structure of *Parp-1^+/+^* mouse incisor. (**d**) Three-dimensional structure of *Parp-1^−/−^* mouse incisor. Hard tissue (enamel and dentin) is drawn in yellow and pulp cavity is drawn in orange. Bar indicates 2 mm.

**Figure 5 cells-08-01157-f005:**
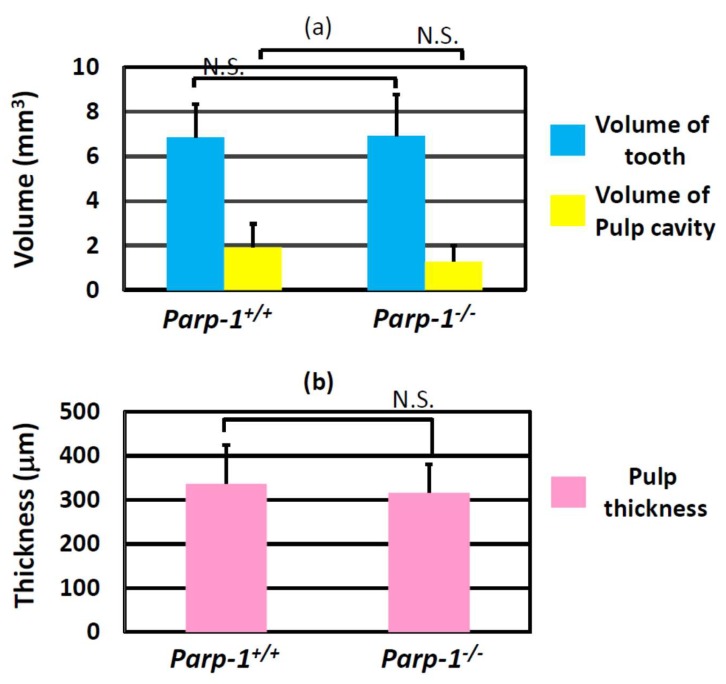
The volume analysis of tooth, pulp cavity, and pulp thickness of incisors of aged *Parp-1^+/+^* and *Parp-1^−/−^* mice (C57BL/6). (**a**) Tooth and pulp cavity volumes of incisors of *Parp-1^+/+^* and *Parp-1^−/−^* mice. (**b**) Pulp thickness of *Parp-1^+/+^* and *Parp-1^−/−^* mice. Six *Parp-1^+/+^* mice and ten *Parp-1^−/−^* mice were used in this analysis. There were no statistically significant differences in the volume of tooth, pulp cavity, and pulp thickness between the genotypes. However, pulp cavity of *Parp-1^−/−^* mice showed a tendency of slightly smaller volume. (*p* = 0.181). N.S., not statistically significant.

**Figure 6 cells-08-01157-f006:**
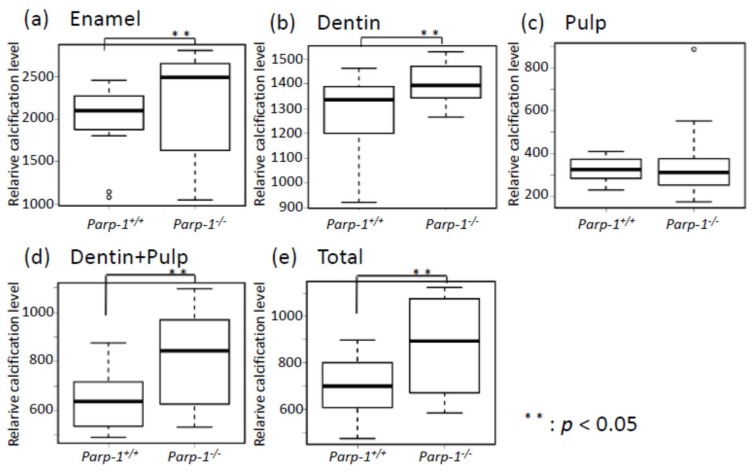
Relative evaluation of calcification levels of each element of tooth in aged *Parp-1^+/+^* and *Parp-1^−/−^* incisors. (**a**) Comparison of relative calcification levels of enamel in *Parp-1^+/+^* and *Parp-1^−/−^*, (**b**) dentin, (**c**) pulp, (**d**) dentin + pulp, and (**e**) total of the tooth. Air was set to be the zero reference. *Parp-1^−/−^* incisors showed significantly higher calcification levels in enamel, dentin, dentin + pulp, and total of tooth. Six *Parp-1^+/+^* mice and ten *Parp-1^−/−^* mice were used in this analysis.

**Figure 7 cells-08-01157-f007:**
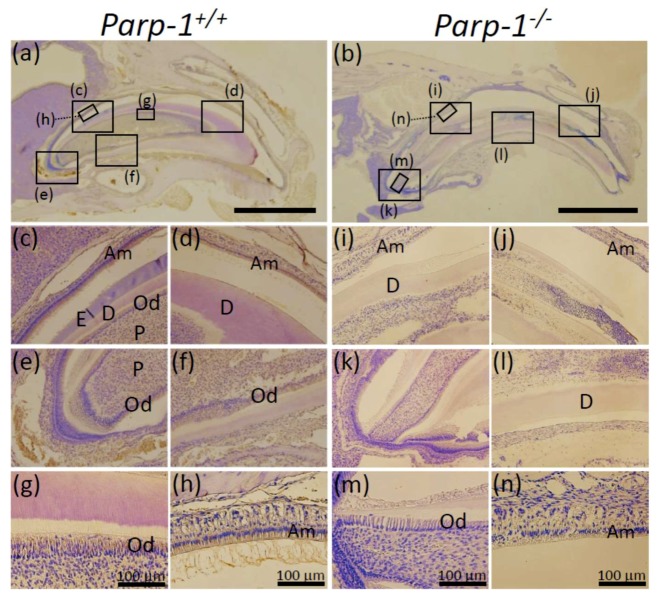
Immunohistochemical analysis of Parp-1 expression in aged *Parp-1^+/+^* and *Parp-1^−/−^* incisors (C57BL/6). Expression of Parp-1 protein in the sagittal plane of *Parp-1^+/+^* (**a**) and *Parp-1^−/−^* incisors (**b**). Parp-1 was detected in odontoblasts (**c,e,f,g**) and ameloblasts (**c,d,h**) of *Parp-1^+/+^* incisor mostly in cytoplasm and occasionally in nuclei, whereas, positive immunostaining was not detectable in odontoblasts (**m**), ameloblasts (**i,j,n**), pulp (**k**), and dentin (**i,l**) of *Parp-1^−/−^* incisor. Semi-quantitative analysis showed that the average values of relative intensities of DAB staining in odontoblast area were 78.2 ± 8.0 and 26.5 ± 7.7 in *Parp-1^+/+^* and *Parp-1^−/−^* incisors, respectively (**g****,m**), whereas that in ameloblast area were 74.6 ± 15.3 and 40.7 ± 3.7 in *Parp-1^+/+^* and *Parp-1^−/−^* incisors, respectively (**h****,n**). Am, ameloblast; Od, odontoblast; E, enamel; D, dentin; P, pulp. Bars of (**a****,b**) indicate 2 mm. The representative results of a mouse for each genotype are shown.

**Figure 8 cells-08-01157-f008:**
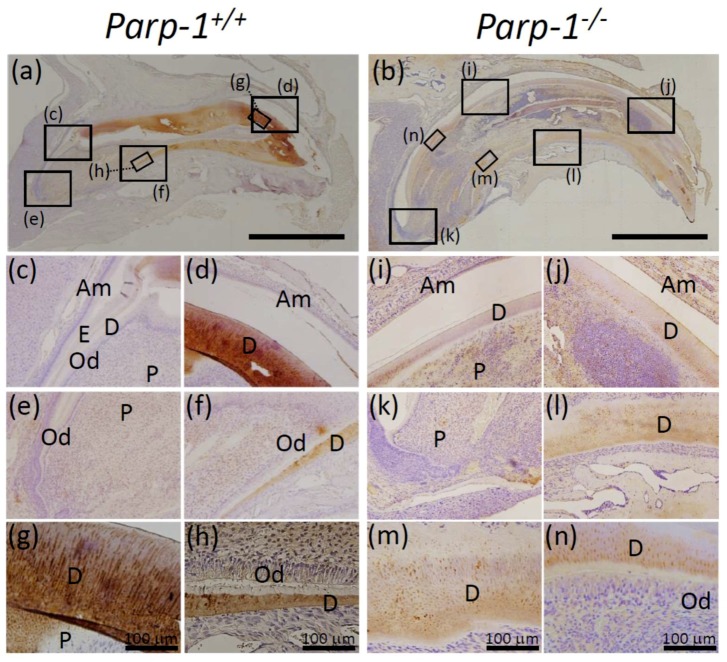
Immunohistochemical analysis of dentin sialoprotein (DSP) expression in aged *Parp-1^+/+^* and *Parp-1^−/−^* incisors (C57BL/6). The expression of DSP protein in the sagittal plane of *Parp-1^+/+^* (**a**) and *Parp-1^−/−^* incisors (**b**). Ameroblasts, odontoblasts, pulp and enamel were not reactive to DSP in *Parp-1^+/+^* incisors (**c**,**e**). *Parp-1^+/+^* incisor showed thickly stained DSP at dentin area (**d**,**f**,**g**,**h**), whereas DSP was sparsely stained at the corresponding area in *Parp-1^−/−^* incisor (**i**–**n**). Moreover, dentinal tubules were clearly observed in dentin area in *Parp-1^+/+^* incisors (**g**), however, they were hardly observed in *Parp-1^−/−^* incisor (**m, n**). Semi-quantitative analysis showed that the average values of relative intensities of DSP positivity in dentin area were 140.6 ± 10.9 and 59.1 ± 5.6 in *Parp-1^+/+^* and *Parp-1^−/−^* incisors, respectively (**g**,**h**,**m**,**n**). Am, ameloblast; Od = odontoblast; E = enamel; D = dentin; P = pulp. Bars of (**a**,**b**) indicate 2 mm. The representative results of a mouse for each genotype are shown.

**Figure 9 cells-08-01157-f009:**
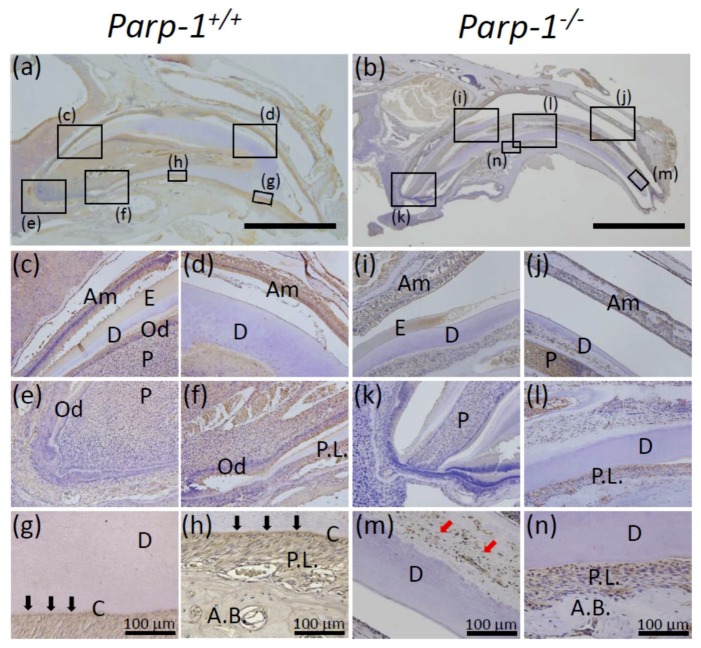
Immunohistochemical analysis of bone sialoprotein (BSP) expression of aged *Parp-1^+/+^* and *Parp-1^−/−^* incisors (C57BL/6). Expression of BSP in the sagittal plane of *Parp-1^+/+^* (**a**) and *Parp-1^−/−^* incisors (**b**). BSP was detected at periodontal ligament in both *Parp-1^+/+^* (**a,f,g,h** ) and *Parp-1^−/−^* incisors (**b**,**l**,**n**). BSP was also detected in dentin area of *Parp-1^−/−^* incisor (**i**–**n**; red arrows) but not in *Parp-1^+/+^* incisor (**a**,**c**,**d**,**e**,**g**). Moreover, BSP positive cementum was observed as linear structure in *Parp-1^+/+^* incisor between dentin and periodontal ligament (**g**,**h**; black arrows), whereas it was hardly observed in *Parp-1^−/−^* incisor (**n**). Semi-quantitative analysis showed that the average values of relative intensities of BSP positivity in dentin area were 42.4 ± 2.7 and 107.8 ± 7.4 in *Parp-1^+/+^* and *Parp-1^−/−^* incisors, respectively (**g**,**m**). Am, ameloblast; Od, odontoblast; E, enamel; D, dentin; P.L., periodontal ligament; P, pulp; A.B., alveolar bone. Arrows indicate the predentin area showing scattered BSP-positive area. Bars of (**a**,**b**) indicate 2 mm. The representative results of a mouse for each genotype are shown.

**Table 1 cells-08-01157-t001:** Incidence of dental dysplasia of aged ICR/129Sv mice.

Genotype	No. of Animals	Incidence (%)		
		Dysplasia	Denticle	Both Dysplasia & Denticle
*Parp-1^+/+^*	13	4 (30.8)	4 (30.8)	4 (30.8)
*Parp-1^−/−^*	9	9 (100.0) *	8 (88.9) *	8 (88.9) *

^*^: *p* < 0.01.

**Table 2 cells-08-01157-t002:** Incidence of dental dysplasia of aged C57BL/6 mice.

Genotype	No. of Animals	Incidence (%)		
		Dysplasia	Denticle	Both Dysplasia & Denticle
*Parp-1^+/+^*	6	2 (33.3)	0 (0.0)	0 (0.0)
*Parp-1^−/−^*	10	9 (90.0) *	4 (40.0) *	4 (40.0) *

*: *p* < 0.01.

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
