# Peer review of "Spontaneous Development of Dental Dysplasia in Aged Parp-1 Knockout Mice"

_cells, 2019, doi:10.3390/cells8101157_

Round 1
Reviewer 1 Report
Hisako Fujihara and her colleagues describe a very interesting phenomenon; the dental dysplasia of aged PARP-1 KO animals. They experienced high prevalence of denticle formation and dental dysplasia in these animals according to histological and CT analysis. On the other hand, the documentation of the results should be substantially improved. The background, the brightness and the contrast of the histological images are continuously changing. The brown color of DAB can be hardly seen in the immunohistochemistry images. The evaluation of the immunohistochemical stainings are not shown. The scale bars are confusing; 2mm in fig 3/f,j might not match with 500µm in fig.2/a,b. Neither the form of data presentation (in the figure legends), nor the used statistical probes are mentioned.
They showed increased calcification and decreased BSP immunpositivity in PARP-1 KO mice, however they failed to explain the connection between these changes and dental dysplasia or PARP-1 expression. It is strange that wild type animals do not show nuclear PARP-1 positivity in any of the examined cell types. It is also contradictory that the average pulp thickness tends to be smaller in PARP KO animals, however their pulp cavities do not narrow toward the incisor margin.
Author Response
We greatly appreciate the constructive comments and advice and edited the manuscript accordingly. The point-by-point responses to the suggestions from the Editor and reviewers are described below. We included the tracked manuscript (with yellow color) that indicates changes and editing.
We hope that our revision will be considered suitable for publication in Cells.
1. The background, the brightness and the contrast of the histological images are continuously changing. The brown color of DAB can be hardly seen in the immunohistochemistry images.
Response: Thank you very much for your comment. We have adjusted brightness and contrast of histological images as much as possible. Moreover, to show the Parp-1 expression of odontoblasts and ameloblasts, we added the magnified images in Figure 7g,h,m,and n, Figure 8 g, h, m, and n, and Figure 9 g, h, m, and n.
2. The evaluation of the immunohistochemical stainings are not shown. The scale bars are confusing; 2mm in fig 3/f,j might not match with 500µm in fig.2/a,b. Neither the form of data presentation (in the figure legends), nor the used statistical probes are mentioned.
Response: As pointed out by the reviewer, scale bars were corrected in Figures 2, 7, 8 and 9. We also described the evaluation results of the immunohistochemical stainings in the legends of Figures 7, 8, & 9 as follows:
“310-312; Parp-1 was detected in odontoblasts (c, d, g) and ameloblasts (e, f, h) of Parp-1+/+ incisors mostly in cytoplasm and occasionally in nuclei, whereas, the positive immunostaining was not detectable in odontoblasts (m) and ameloblasts (n) of Parp-1-/- incisors.
329-332; Parp-1+/+ incisors showed thickly stained DSP at dentin area (d, f, g, h), whereas DSP was sparsely stained at the corresponding area in Parp-1-/- incisors (i, j, k, l). Moreover, dentinal tubules were clearly observed in dentin area in Parp-1+/+ incisors (g), however, they were hardly observed in Parp-1-/- incisors (m, n).
347-351; BSP was detected at periodontal ligament in both Parp-1+/+ (c, d-h) and Parp-1-/- incisors (i-n). BSP was also detected in dentin area of Parp-1-/- incisors (m, red arrows) but not in Parp-1+/+ incisors (a, d, g). Moreover, BSP positive cementum was observed as linear structure in Parp-1+/+ incisors between dentin and periodontal ligament (g, h, black arrows), whereas it was hardly observed in Parp-1-/- incisors (n).
3. They showed increased calcification and decreased BSP immune-positivity in PARP-1 KO mice, however they failed to explain the connection between these changes and dental dysplasia or PARP-1 expression.
Response: Thank you very much for this comment. The decreased DSP positivity and occult BSP positivity observed in Parp-1-/- mice incisors is one of striking results of our manuscript. We added the explanation for the relationship between these changes and Parp-1 expression in line 421-427 as follows: “Therefore, the possible mechanism of Parp-1 involvement in odontogenic development of incisors at an advanced age could be 1) Parp-1 might be directly involved in regulation of DSP expression in odontoblasts, 2) the loss of Parp-1 expression might have caused a failure in differentiation from ectomesenchymal cells into odontoblasts in Parp-1-/- incisors, which resulted in increased differentiation into osteoblasts, 3) the layer formation of odontoblasts was also affected by the loss of Parp-1 expression. As a result, relative calcification levels of enamel and dentin would become significantly higher in Parp-1-/- incisors.”
4. It is strange that wild type animals do not show nuclear PARP-1 positivity in any of the examined cell types.
Response: According to your kind comments, we re-analyzed the immunostaining at magnified scale and we found occasional staining of Parp1 in nuclei as well as major staining in cytoplasm. We consider the long term decalcification may have caused the preferential loss of antigen detection in nuclei compared with cytoplasm. Therefore, we corrected the description as follows;
300-304.: “Parp-1+/+ incisors showed positive Parp-1 immunostaining in ameloblast (7c, h) and odontoblast (7f, g) layers and their localization was mostly in cytoplasm but occasionally in nuclei (Figure 7a, c, d, e, f, g, h). The long term decalcification process may have caused the preferential loss of antigen detection in nuclei compared with cytoplasm.”
418-427; “Parp-1 is known as nuclear as well as cytoplasmic protein [32] and it is occasionally detected in cytoplasm in tissue staining [33]. We consider that Parp-1 activity either in nuclei or cytoplasm of Parp-1+/+ mice could contribute as a possible differentiation regulator for odontoblasts and ameloblasts. Therefore, the possible mechanism of Parp-1 involvement in odontogenic development of incisors at an advanced age could be 1) Parp-1 might be directly involved in regulation of DSP expression in odontoblasts, 2) the loss of Parp-1 expression might have caused a failure in differentiation from ectomesenchymal cells into odontoblasts in Parp-1-/- incisors, which resulted in increased differentiation into osteoblasts, 3) the layer formation of odontoblasts was also affected by the loss of Parp-1 expression. As a result, relative calcification levels of enamel and dentin would become significantly higher in Parp-1-/- incisors.”
5. It is also contradictory that the average pulp thickness tends to be smaller in PARP KO animals, however their pulp cavities do not narrow toward the incisor margin.
Response: Thank you very much for your comment. The mice used in Figures 3 and 4 are same. Although they are representative ones, Parp-1-/- mice showed varied pulp thickness of pulp cavities and some showed almost closed pulp cavity inside of incisors. As a result of this, pulp thickness tends to be smaller in Parp-1-/- mice incisors. We have added this explanation as follows in line 392-400.
“the parameters of the general structure of incisors at an advanced age, including volume of the tooth, that of pulp cavity and thickness of pulp, did not exhibit statistically significant differences between either genotype, and only pulp cavity showed a tendency to be smaller in Parp-1-/- incisors. The reason why the parameters of the general structure did not show significant difference could be the wide range of morphological variations in incisors of Parp-1-/- mice; Parp-1-/- incisors showed varied thickness of pulp based on the diverse shapes of pulp cavities, for example, some of them showed almost a closed pulp cavity because of the denticle structure inside of the incisors. As a result of this, pulp thickness seems to be larger in Parp-1-/- incisors (Figure 3a-d and 4c,d), however, there was a tendency for a smaller pulp cavity in Parp-1-/- incisors (p = 0.181, Figure 5a).”
Reviewer 2 Report
In the paper, Fujihara et al. demonstrated that aged Parp-1 mouse incisor showed dental dysplasia and denticle formation in both ICR/129sv and C57BL/strain. Although the research is design appropriate, several issues exist in the introduction, discussion and in the presentations of results. So I suggest that the paper should be reconsider after major revision.
First of all, the Introduction and the discussion need to be improved avoiding repetitions, often not clear. In particular, by comparing the lines 59-66 with 383-386, and 76-82 with 394-399. It is therefore necessary to remove the repetitions and the ambiguity. In fact, the authors present the results for both ICR/129sv and C57BL/strain, but on ICR/129sv they did only histopatology analys so for me these results are incomplete, and in the discussion this issue must be clarify.
In Materials and methods, the authors described to use only male mice, it is important to repeat this during the discussion of results and the explanation of experiments. in particular, in the paragraph 3.5 the authors didn’t find significant differences between genotype differences in either tooth, pulp volume or in the pulp thickness, probably the result could be different for female mice (see Morgan Goss, Molecular Genetics and Metabolics (2019)).
I suggest to modify the Figure 3g-j, the authors could be renaming it to Fig 4a-d and consequently rename the next figures, or bring together Fig. 3a-f with Fig. 3g-j. Moreover, in the Fig3 a-f specify the color bar used for the gray scale.
Figures 4 and 5: put labels on y-axis.
Finally, line 325, probably there is a mistake with the letter of figure 8f-j.
Author Response
We greatly appreciate the constructive comments and advice wers and edited the manuscript accordingly. The point-by-point responses to the suggestions from the Editor and reviewers are described below. We included the tracked manuscript (with yellow color) that indicates changes and editing.
We hope that our revision will be considered suitable for publication in Cells.
1. First of all, the Introduction and the discussion need to be improved avoiding repetitions, often not clear. In particular, by comparing the lines 59-66 with 383-386, and 76-82 with 394-399. It is therefore necessary to remove the repetitions and the ambiguity.
Response: Thank you for your comment. According to your valuable instruction, the introduction and the discussion section have been drastically modified and all repetitions were removed. And we also tried removing the ambiguity and letting them simpler.
2. In fact, the authors present the results for both ICR/129sv and C57BL/strain, but on ICR/129sv they did only histopathology analysis so for me these results are incomplete, and in the discussion this issue must be clarify.
Response: We tried immunostaining analysis of ICR/129Sv strain using antibodies against Parp-1, DSP and BSP, however, the signals were not detected with the antibodies, possibly because the samples of ICR/129Sv mice were relatively older and not in good condition after long-term decalcification process. Therefore, we added the explanation as follows in line 362-366.
“We analyzed aged Parp-1+/+ and Parp-1-/- incisors of both an ICR/129Sv mixed genetic background and C57BL/6 strain histopathologically but immunohistochemical analysis for incisors of ICR/129Sv mixed genetic background was not successiful using anti-Parp-1, DSP and BSP antibodies, possibly because the incisor samples of ICR/129Sv mixed genetic background were relatively older and not in good condition after long-term decalcification process.”
3. In Materials and methods, the authors described to use only male mice, it is important to repeat this during the discussion of results and the explanation of experiments. in particular, in the paragraph 3.5 the authors didn’t find significant differences between genotype differences in either tooth, pulp volume or in the pulp thickness, probably the result could be different for female mice (see Morgan Goss, Molecular Genetics and Metabolics (2019)).
Response: Thank you very much for your kind comment. Although we used only male mice to avoid possible gender difference for dental dysplasia. We added that description of the phenomena was evaluated only in male and female were not analyzed and a possible gender difference should be studied in the further study.
We added the suggested article by the reviewer as a new reference and added new comments as follows in line 395-411. Moreover, the CT photos of right incisors were added as Figure 3 (g) and (h).
“The reason why the parameters of the general structure did not show significant difference could be the wide range of morphological variations in incisors of Parp-1-/- mice; Parp-1-/- incisors showed varied thickness of pulp based on the diverse shapes of pulp cavities, for example, some of them showed almost a closed pulp cavity because of the denticle structure inside of the incisors. As a result of this, pulp thickness seems to be larger in Parp-1-/- incisors (Figure 3a-d and 4c,d), however, there was a tendency for a smaller pulp cavity in Parp-1-/- incisors (p = 0.181, Figure 5a). In this study, only male mice were analyzed because it was preferred to analyze the effect of Parp-1 deficiency on dental differentiation at an advanced age in the absence of the effect of the estrous cycle. However, a gender difference of tooth formation has been reported using Trps1+/- (trichorhinophalangeal syndrome heterozygous) mice showing significantly smaller crown and root volumes in female Trps1+/- mice compared with male Trps1+/- mice [30]. Trichorhinophalangeal syndrome is a skeletal dysplasia with skeletal defects as well as dental abnormalities, where Trps1 gene regulates dental mineralization. Although Parp-1 deficiency has not been reported to be associated with such autosomal dominant genetic disorders, it could be possible that aged female Parp-1-/- incisors might show a difference from male ones in their general structure, considering that dental mineralization could be affected by gender. Therefore, possible gender differences should be studied in further.”
4. I suggest to modify the Figure 3g-j, the authors could be renaming it to Fig 4a-d and consequently rename the next figures, or bring together Fig. 3a-f with Fig. 3g-j. Moreover, in the Fig3 a-f specify the color bar used for the gray scale.
Response: Thank you for your suggestion. We renamed Figure 3g-j to Figure 4a-d, and renamed followed figures. Moreover, the color bar used in Figure 3 had been changed its color to same one in Figure 4.
5. Figures 4 and 5: put labels on y-axis.
Response: Thank you for your comment. The labels were put in Figures 5 and 6 (former Figure 4 and 5) in each panel.
6. Finally, line 325, probably there is a mistake with the letter of figure 8f-j.
Response: Thank you for your comment. It was corrected to Figure 9b, l, n (former Figure 8b,j).
Reviewer 3 Report
The manuscript by Hisako Fujihara et al., reports that Parp-1 deficiency could be involved in development of dental dysplasia at an advanced age. In addition to that, these studies show that Parp-1 is not essential in dentinogenesis during development although it is possibly involved in the regulation of continuous dentinogenesis of incisors.
The findings are sufficient for publication on “Cell MPDI” after revision
Major Comments
1. The Authors could combine paragraph 1 and 2, it is unnecessary to keep them separate.
2. To give value to the methods the Authors could insert some references to the applied methodologies.
3. In Figure 3, the Authors did not include information on the size of the bar.
4. In Figure 4 and legend, there are not statistical indications.
Author Response
We greatly appreciate the constructive comments and advice wers and edited the manuscript accordingly. The point-by-point responses to the suggestions from the Editor and reviewers are described below. We included the tracked manuscript (with yellow color) that indicates changes and editing.
We hope that our revision will be considered suitable for publication in Cells.
1. The Authors could combine paragraph 1 and 2, it is unnecessary to keep them separate.
Response: Thank you very much for your comment. The paragraph 1 and 2 were combined.
2. To give value to the methods the Authors could insert some references to the applied methodologies.
Response: Thank you very much for your comment. Two references as follows were cited to section of Material and Methods.
27. Koda, N.; Sato, T.; Shinohara, M.; Ichinose, S.; Ito, Y.; Nakamichi, R.; Kayama, T.; Kataoka, K.; Suzuki, H.; Moriyama, K.; Asahara, H. The transcription factor mohawk homeobox regulates homeostasis of the periodontal ligament. Development 2017, 144, 313–320, doi:10.1242/dev.135798.
28. Yasukawa, M.; Fujihara, H.; Fujimori, H.; Kawaguchi, K.; Yamada, H.; Nakayama, R.; Yamamoto, N.; Kishi, Y.; Hamada, Y.; Masutani, M. Synergetic Effects of PARP Inhibitor AZD2281 and Cisplatin in Oral Squamous Cell Carcinoma in Vitro and in Vivo. International journal of molecular sciences 2016, 17, 272, doi:10.3390/ijms17030272.
3. In Figure 3, the Authors did not include information on the size of the bar.
Response: Thank you very much for your comment. The size of bar was added in Figure 3.
4. In Figure 4 and legend, there are not statistical indications.
Response: Thank you very much for your comment. The statistical indication of “N.S. = not significant” was added in Figure 5 (former Figure 4) and also added in the legend.
Round 2
Reviewer 1 Report
The authors improved the manuscript and answered the raised questions; however, there are still concerns that should be addressed.
1. Although they described the evaluation of the immunohistochemical stainings, it still lacks any concrete results. The number of cases should be mentioned for all stainings. They should be evaluated in a semi-quantitative way (score system) or with a computer-based method (ImageJ/Fiji is a freely available software for these purposes).
2. Used statistical methods should be described. They possibly used khi-square test in tables 1 and 2; and Students’ t-test or Mann-Whitney U-test in figures 5 and 6.
3. In figure 6, they use two stars (**) for p<0.05; however, by convention one star (*) should be used.
4. The BSP staining of the dentin area of PARP-/- incisors are also showing cells in this area, and these are the cells that show positivity. What the authors think these cells are?
Author Response
1. Although they described the evaluation of the immunohistochemical stainings, it still lacks any concrete results. The number of cases should be mentioned for all stainings. They should be evaluated in a semi-quantitative way (score system) or with a computer-based method (ImageJ/Fiji is a freely available software for these purposes).
Response: Thank you very much for your comment. The numbers of cases analyzed histopathologically was four Parp-1-/- and four Parp-1+/+ mice and the number of mice radiologically analyzed were Six Parp-1+/+ and ten Parp-1-/- mice. Therefore, these numbers were described in the line 114 and 152 as follows.
line 114: Six Parp-1+/+ and ten Parp-1-/- mice were used in this analysis.
line 152: Four Parp-1+/+ and four Parp-1-/- mice were used in this analysis.
Regarding semi-quantitative evaluation, we tried to analyze using ImageJ/Fiji according to your instruction. All photographs were deconvoluted and DAB color version images were chosen from divided three images. Then, subjected area was selected and mean density of DAB color was calculated and the number was converted from max intensity 255. This analysis was performed only one case of each genotype due to the condition of sample on slides, which were not as good as usual IHC analysis. The samples on slides tend to be peeled off during IHC procedure and it is probably because of long-term formalin fixation (more than 10 years) and decalcification process.
Moreover, positive inner control for IHC procedure is difficult to be chosen because Parp-1 KO mice do not have positive internal control for IHC. Therefore, the result of deconvluted DAB color was quantified and compared with back ground resulting in 2.95-fold-strength in PARP-1 positivity in odontoblast area of Parp-1+/+ incisors and 1.83-fold-strength in ameloblast area of Parp-1+/+ incisors, 2.38-fold-strength of DSP in dentin area of Parp-1+/+ incisors. The strength of BSP positivity was 2.54-fold-strength in dentin area of Parp-1-/- incisors, respectively. Therefore, methods and results of these quantitative analysis was described as follows:
line 155-164:
2.6. Quantification of intensity of immunohistochemical positivity
The photographs of the immunohistochemical analysis with each antibody were analyzed by ImageJ Fiji (National Institutes of Health, Bethesda, MD, USA, available form https://imagej.net/Fiji/Downloads) and mean intensity of selected area was calculated. First, all photographs were deconvoluted and DAB color version images were chosen from divided three images. Then, subjected area was selected and cropped; 1) odontoblasts lineage for Parp-1 positivity, 2) ameloblast lineage for Parp-1 positivity, 3) dentin area for DSP positivity, and 4) dentin area for BSP positivity. Subsequently, mean density of DAB color was calculated and the number was converted from max intensity 255 [29]. The representative immunostaining data for Figures 7,8 and 9 of each genotype was used in this analysis.
line 325-328:
To quantify the intensity of DAB staining for Parp-1 staining, photos of both Parp-1+/+ and Parp-1-/- incisors represented in Figure 7 were analyzed by ImageJ Fiji. The average value of relative intensity of DAB staining of Parp-1 was 2.95 folds higher in odontoblast area (Figure 7g, m) and 1.83 folds higher in ameloblast area compared with Parp-1+/+ incisors (Figure 7h, n), respectively.
line 346-349:
To quantify the intensity of DAB staining for DSP staining in dentin area, photos of both Parp-1+/+ and Parp-1-/- incisors represented in Figure 8 were analyzed by ImageJ Fiji. The average value of relative intensity of DSP positivity in dentin area was 2.38 folds higher in Parp-1+/+ incisors compared with Parp-1-/- incisors (Figure 8h, n).
line 368-371:
To quantify the intensity of DAB staining for BSP staining in dentin area, photos of both Parp-1+/+ and Parp-1-/- incisors represented in Figure 9 were analyzed by ImageJ Fiji. The average value of relative intensity of BSP positivity in dentin area was 2.54 folds higher in Parp-1-/- incisors compared with Parp-1+/+ incisors (Figure 9g, m).
2. Used statistical methods should be described. They possibly used khi-square test in tables 1 and 2; and Students’ t-test or Mann-Whitney U-test in figures 5 and 6.
Response: Thank you very much for your comment. The detail of statistical method was added in line 166-173 as follows.
2.7. Statistical analysis
All statistical analyses were performed with EZR (Saitama Medical Center, Jichi Medical University, Saitama Japan), which is a graphical user interface for R (The R Foundation for Statistical Computing, Vienna, Austria). More precisely, it is a modified version of R Commander designed to add statistical functions frequently used in biostatistics [30]. In detail, chi square test was performed for the analysis of the incidence of dysplasia and denticle in incisors of both genotypes (Tables 1 and 2). Students’ t-test was performed for the analysis of general structure of incisors of both genotypes (Figures 5 and 6).
3. In figure 6, they use two stars (**) for p<0.05; however, by convention one star (*) should be used.
Response: Thank you very much for your comment. (*) was applied for p<0.05.
4. The BSP staining of the dentin area of PARP-/- incisors are also showing cells in this area, and these are the cells that show positivity. What the authors think these cells are?
Response: Thank you very much for your comment. As you mentioned, cell-like structures of Parp-1-/- incisors are BSP positive. Generally, dentin and predentin do not have cells in their matrix although Parp-1-/- incisors show possible cell-like structures which are positive to BSP. Perhaps, these cells could be osteoblasts or osteoblast-like cells because they are BSP-positive but could not be odontoblasts. To confirm what they are, we also performed IHC of osterix, osteocalcin and runx2 in addition to BSP, however, our samples of both genotypes were not reactive to these markers except BSP. Probably it is due to the same reason of previous issue - they were fixed in formalin for years and they were also decalcified. Therefore, these cells have not been fully characterized yet, and they still could be other cell types.
Considering that 1) odontoblasts are not reactive to BSP, whereas they are reactive to DSP, and 2) osteocytes are usually observed in cortical bone with the structure of Osteon, cells observed in the dentin matrix in Figure 9 may be osteoblast or osteoblast-like cells. At this moment, we think two hypotheses about why osteoblasts could be observed in dentin; 1) they are mistakenly put in the line to osteoblasts which were differentiated from precursor cells in dental pulp, which should have become odontoblast, because of Parp-1 deficiency, 2) osteoblasts or osteoblast-like cells were migrated from somewhere (perhaps migration from bone marrow) considering Parp-1 KO mice showed ectopic bone formation in liver (in preparation for submission).
Therefore, further investigation would be necessary to further understand how Parp-1 activity is involved in dentinogenesis/odontogenesis during aging including characterization of these BSP-positive cells in dentin area.
Reviewer 2 Report
Thank you for your revised version. I believe that the present form Is ready for pubblication.
Author Response
Thank you for your revised version. I believe that the present form Is ready for publication.
Response: Thank you very much for your comment. It is very grateful for your decision. And the manuscript was English edited again for this secondary submission.